# Charcoal morphologies and morphometrics of a Eurasian grass-dominated system for robust interpretation of past fuel and fire type

**Angelica Feurdean**[1,2]**, Richard S. Vachula**[3]**, Diana Hanganu**[4,5]**, Astrid Stobbe**[6]**, and Maren Gumnior**[6]

[1]Department of Physical Geography, Goethe University, Altenhöferallee 1, 60438 Frankfurt am Main, Germany
[2]STAR-UBB Institute, Babeş-Bolyai University, Kogălniceanu 1, 400084, Cluj-Napoca, Romania
[3]Department of Geosciences, Auburn University, 2070 Beard Eaves Coliseum, AL, USA `CE1`
[4]ICUB Research Institute of the University of Bucharest, Panduri 90–92, Bucharest, Romania
[5]Emil Racoviță Institute of Speleology, Frumoasă 31, 010986, Bucharest, Romania
[6]Archaeobotanical Laboratory, Institute for Archaeological Sciences, Goethe University, Norbert-Wollheim-Platz, 160629 Frankfurt am Main, Germany

**Correspondence:** Angelica Feurdean (feurdean@em.uni-frankfurt.de)

**Abstract.** ~~Reconstructing past fire regimes by quantifying charcoal fragments is a commonly used approach, and~~ recent developments in morphological and morphometric analyses of charcoal particles have improved our ability to discern characteristics of burnt plant fuel and interpret fire-type changes. However, burning experiments linking known plants to these metrics are limited, particularly in open ecosystems. This study presents novel analyses of laboratory-produced charcoal of 22 plant species from the steppe regions of Eurasia (Romania and Russia), along with selected samples from three Holocene charcoal and pollen records from the same areas. We characterise charcoal production, morphologies and morphometrics in these grass-dominated environments, thereby enabling more robust interpretations of fuel sources and fire types for palaeofire research. Our experiments demonstrate that fire temperature can introduce biases in charcoal produced ~~depending on~~ species. Grass charcoal production was significantly lower and decreased more strongly with fire temperature compared to forbs. This suggests an underrepresentation of terrestrial graminoids in sedimentary charcoal assemblages. ~~While charcoal morphologies enable finer distinctions between fuel types than morphometrics, both approaches are complementary for fuel identification.~~ Morphometric analyses revealed that graminoid charcoal particles were more elongated (length-to-width ratio $L/W = 4$) and narrower (width-to-length ratio $W/L = 0.38$) than forbs ($L/W = 3.1$ and $W/L = 0.42$, respectively), in agreement with a global compilation for graminoids ($\langle L/W = 4.3$ for grass 5.4 grass and wetland graminoids) and forbs ($L/W = 2.9$). However, overlapping $L/W$ values present a challenge for establishing cut-off values for fuel type identification in charcoal assemblages with mixed fuel sources. Based on our analyses and compiled datasets from experimental burns, $L/W$ values above 3.0 may indicate predominantly herbaceous morphologies in temperate grassland-dominated ecosystems, though values are likely to be higher for grass than forb-dominated ~~vegetation.~~ Notably, terrestrial grasses exhibit shorter aspect ratios (4.3) than wetland graminoids (6.4), highlighting that the aspect ratio needs tailoring to the specific environment of its application, i.e. wetland vs. terrestrial ecosystems. The long forms of graminoid charcoal particles also suggest their potential for atmospheric longer-distance transport compared to more spherical particles, meaning they likely provide insights into regional fire history. An important finding is that charcoal ~~morphology and morphometrics~~ of herbaceous plants closely corresponded to ~~the dominant plant communities indicated by~~ the pollen record, highlighting a solid link between the dominant vegetation ~~type~~ and fuel burnt in grassland-dominated environments. However,

the relationship between woody charcoal and tree pollen may be more complex, as tree pollen can travel atmospherically longer distances compared to woody charcoal. Our results also highlight the complex interplay between local vegetation and charcoal composition with human fire use that needs to be considered when interpreting charcoal morphological records. A critical takeaway from this study is the importance of not assuming the universality of previous research findings and instead employing experimental approaches to characterise charcoal particles in new ecosystems prior to the application of these techniques. Furthermore, this study also highlights recommendations for further research in new geographical areas and proposes methodological adjustments to enhance the usefulness of charcoal analysis in fire research.

## 1   Introduction

Fire plays a crucial role in shaping open ecosystems like grasslands and grass–tree mosaics (i.e. steppe, savanna, forest–steppe, woodlands) at evolutionary to ecological timescales (Bond et al., 2005) TS1. One of the largest extents of open ecosystems is in Eurasia, which has been heavily impacted by human activities for millennia. Consequently, little is known about the natural occurrence and intensity of fires in relation to climate, biomass amount and vegetation composition in these ecosystems (Feurdean et al., 2013, 2021; Leys et al., 2018; Unkelbach et al., 2018; Feurdean and Vasiliev, 2019; Lukanina et al., 2023). Long-term records of wildfire activity are essential for understanding how fire regimes vary with changes in climate, vegetation and human–vegetation interactions (Bowman et al., 2009).

Charcoal, a by-product of the incomplete combustion of plant tissues, is the most widely used proxy for the reconstruction of past fire activity (Whitlock and Larsen, 2002 TS2). A few keys for identifying small charcoal particles in natural sedimentary archives have been made available. They are either based on quantitative characteristics of charcoal size and form (morphometrics such as aspect ratio, surface area, area-to-perimeter ratio) or qualitative morphological characteristics such as edge aspects, surface features, cleavage, lustre, tracheids with border pits, leaf veins, cuticles, etc. (Umbanhowar and McGrath, 1998; Enache and Cumming, 2006 TS3; Jensen et al., 2007; Courtney Mustaphi and Pisaric, 2014; Pereboom et al., 2020; Feurdean, 2021 TS4). These keys can help link charcoal particles to plant traits, such as plant types (moss, graminoids, forbs, shrubs/trees) or plant parts (stems, branches, roots, leaves, wood). Most of the available keys for identifying charcoal morphologies and morphometries have been developed based on the description of sedimentary charcoal from boreal forests, with a few emerging studies focusing on experimentally produced charcoal of known plants or in other ecosystems (Umbanhowar and McGrath, 1998; Jensen et al.,

2007; Crawford and Belcher, 2014; Li et al., 2019 TS5; Pereboom et al., 2020; Feurdean, 2021). As a result, the accuracy and applicability of charcoal morphometries in ecosystems with different dominant plant compositions still need to be determined. Moreover, the threshold values of the aspect ratio, i.e. length-to-width ($L/W$) or width-to-length ($W/L$), of sedimentary charcoal for fuel identification often contrast with those shown by laboratory experiments, highlighting the need for further research (see review by Vachula et al., 2021). Other metrics, such as surface area and area-to-perimeter ratio ($A/P$), have also been explored to describe regular versus irregular charcoal shapes (Lestienne et al., 2020; Pereboom et al., 2020; Feurdean, 2021), but their full potential requires further evaluation. Another uncertainty in charcoal-based fire reconstruction is how charcoal production is affected by the biomass quantity, density, moisture content and fire temperature (Simpson et al., 2016; Pereboom et al., 2020; Feurdean, 2021; Hudspith et al., 2018). Finally, the size, shape and density of the charcoal fragments can cause substantial differences in particle dispersal and incorporation into sediments (Clark and Hussey, 1988 TS6; Vachula and Richter, 2018; Vachula and Rehn, 2023 TS7). Microscopic particles may be transported considerable distances and deposited into various sedimentary environments such as peats, lakes and rivers, in contrast to larger particles that tend to be deposited nearer to the source of the fire (Scott et al. TS8, 2000; Scott, 2010). Additionally, the morphology of charcoal characterised by longer and thinner shapes, typical of grass, allows for longer-distance atmospheric transport compared to spherical or denser particles, such as those from leaves and wood (Clark and Hussey, 1988; Pisaric, 2002 TS9; Scott et al., 2000; Vachula and Richter, 2018; Courtney Mustaphi et al., 2022 TS10; Vachula and Rehn, 2023). Conversely, water favours the transport of wood charcoal (Scott et al., 2000). Thus, a better understanding of the factors influencing charcoal production, morphometries and dispersal is essential for more accurate reconstructions of past fire activity.

This study presents the first analyses of laboratory-produced (muffle oven) charcoal from a variety of grass, forb and shrub taxa from Eurasian steppe to better understand the diversity of charcoal morphologies and morphometries in grassland-dominated ecosystems and facilitate more robust interpretations of fuel sources. By comparing the results to sedimentary charcoal morphologies, morphometrics and pollen assemblages from the same regions, we aim to provide more accurate interpretations of past vegetation, fuel sources and fire types. The study also compares aspect ratio results to a compiled literature dataset to evaluate this metric's universality. Specifically, we determine (i) morphometric and morphological distinction between taxa and plant parts, including photographic plates; (ii) thresholds in charcoal morphometrics such as $L/W$, $W/L$, and $A/P$ ratios indicative of systems dominated by grasslands or a grass–tree mosaic such as steppe, savanna, forest–steppe, and woodlands; and (iii) the effect of combustion temperature on the

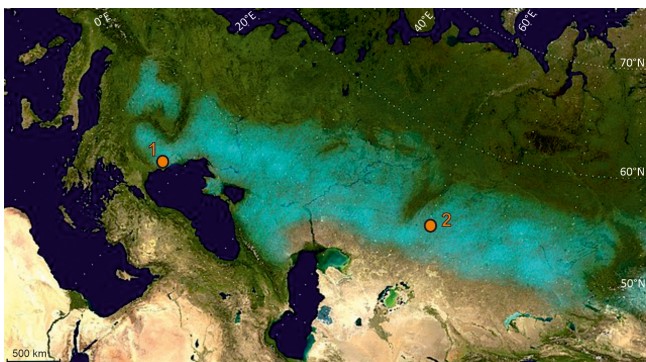

**Figure 1.** The distribution of open ecosystems (steppes/grasslands and forest–steppes) in Eurasia. Mangalia Herghelie, Black Sea, SE Romania (1), and Tom and SH sites, Konoplyanka, Trans-Urals, Russia (2), are located in the steppe ecosystems. The source image is a product of NASA's Blue Marble Project public domain.

charred mass of various taxa relative to the quantity of unburned biomass. Ultimately, our approach aims to provide an accessible tool to understand fire regimes in temperate grass-dominated ecosystems.

## 2   Material and methods

### 2.1   Laboratory analysis

We collected plant specimens from 7 graminoid, 15 forb, and 1 shrub species from the steppe area in the Dobrogea, Black Sea, Romania, and Konoplyanka, Trans-Urals, Russia (Fig. 1; Table 1). These specimens are representative of the most prevalent plant taxa in the study area; however, this taxa list constitutes only a partial representation of common grasses and forbs. Many of these species are distributed in small patches due to the extensive conversion of natural steppe into agricultural fields. Tree patches of *Quercus* and *Carpinus* occur in Dobrogea, and *Betula* and *Salix* are present near the Russian sites. However, at the Romanian sites, the forest zone (thermophilous broadleaved) is closer (ca. 100 km) than for the sites (*Pinus–Betula*) in Russia (150 km).

To determine the effect of increasing temperatures on the mass, morphometric characteristics and morphologies of charred plant material, we dried these plants in a desiccator (40 °C) for 24 h. Subsequently, following the protocol of Feurdean (2021), we roasted the plant material in a muffle furnace at five district temperature settings: 250, 300, 350, 400 and 450 °C. For each temperature subset, remains of individual plant species from the entire plant or as selected plant parts (Table 1) were placed in ceramic crucibles, weighed, then placed into the cold muffle oven covered with a lid to limit oxygen availability and avoid mixing the charred particles. The temperature was gradually raised for 1 h, af-

ter which it was held constant for a duration of 2 h. Crucibles with charred plants were cooled in a desiccator, then weighed to calculate the charred to pre-burning mass ratio. A small part of the charred mass was gently disaggregated with a mortar and pestle to mimic the natural breakage that charcoal particles experience over time through sedimentation processes (Umbanhowar and McGrath, 1998; Crawford and Belcher, 2014; Belcher et al., 2015). The charred and disaggregated sample was then washed through a 125 μm sieve to remove smaller fragments. This charcoal size fraction ($> 125$ μm) is directly comparable to the size fraction analysed by the bulk of palaeofire studies (Vachula, 2018). Photographs of charcoal particles were manually taken at $4\times$ magnification with a digital camera (KERN ODC 241 tablet camera). The charcoal particles and morphometric measurements, including the major ($L$) and minor ($W$) axes surface area ($A$), and perimeter ($P$) for individual charred particles larger than 150 μm, were automatically determined from these photographs using the algorithm of Feurdean (2021). Subsequently, we calculated the aspect ($L/W$; $W/L$) and $A/P$ ratios. These metrics were applied on more than 150 charcoal particles (range between 41 and 508), per sample; the lower number of measurements generally corresponds to samples burnt at high temperatures, where particles were more susceptible to breakage or partial ashing (Table S1 in the Supplement; Feurdean, 2023). A limited number of charcoal particles were excluded from these quantifications because they could not be adequately identified. This misclassification was due to their blurred aspect, partial overlapping particles or particles positioned partially outside of the photograph frame. Finer diagnostic features such as shape and surface features (e.g. reticulates, leaf veins), the arrangement of epidermal cells, and cuticles with stomata were characterised by observing the charred particles under a stereomicroscope.

To evaluate the comparability of our aspect ratio values with published experimental charcoal data derived from known plant taxa, we used a global dataset compiled by Vachula et al. (2021), supplemented by data collected from boreal Siberia (Feurdean, 2021). We classified the mean $L/W$ values into distinct growth forms, which included dry grasses ($n = 16$), wetland graminoids ($n = 12$), forbs ($n = 24$), as well as plant parts, encompassing leaves ($n = 33$) and wood ($n = 31$). To assess how the $L/W$ values of Eurasian grasslands compare with global values, we aggregated growth forms and plant parts separately within Eurasian grasslands and the global dataset. To identify a potential threshold aspect ratio value that could distinguish grass or grass–tree-dominated vegetation, we assembled $L/W$ values of charcoal derived from grasses and forbs into the non-woody category and those from tree trunks and branches into woody types. Where available, we included error ranges for the $L/W$ values of the experimental charcoal; however, not all publications provided these values. To visu-

**Table 1.** List of plants burned from Dobrogea, the Black Sea, Romania (RO), and Konoplyanka, Trans-Urals, Russia (RU). All grasses are $C_3$.

| Plant type | Scientific name | Family | Common name | Plant burned |
|------------|-----------------|--------|-------------|--------------|
| Grass | | | | |
| Grass_RO | *Agropyron repens* | Poaceae | Couch grass | Stem/leaves |
| Grass_RO | *Agropyron repens* | Poaceae | Couch grass | Inflorescence |
| Grass_RO | *Agropyron cristatum* | Poaceae | Crested wheatgrass | Stem/leaves |
| Grass_RO | *Agropyron cristatum* | Poaceae | Crested wheatgrass | Inflorescence |
| Grass_RO | *Agropyron* sp. | Poaceae | Wheatgrass | Stem/leaves |
| Grass_RO | *Festuca* sp. | Poaceae | Creeping bent grass | Stem/leaves |
| Grass_RO | *Poa* sp. | Poaceae | Meadow–grass | Stem/leaves |
| Grass_RO | *Bromus sterilis/tectorum* | Poaceae | Bromegrass | Stem/leaves |
| Grass_RO | *Stipa capillata* CE2 | Poaceae | ~~Feder~~ grass CE3 | Whole |
| Forbs | | | | |
| Forb_RO | *Anthemis tinctoria* | Asteraceae | Golden marguerite | Stem/leaves |
| Forb_RO | *Anthemis tinctoria* | Asteraceae | Golden marguerite | Inflorescence |
| Forb_RO | *Artemisia campestris* | Asteraceae | Field wormwood | Whole |
| Forb_RO | *Xeranthemum annuum* | Asteraceae | Annual everlasting | Whole |
| Forb_RO | *Achillea* sp. | Asteraceae | Yarrows | Whole |
| Forb_RO | *Leonurus* sp. | Lamiaceae | | Whole |
| Forb_RO | *Eryngium campestre* | Apiaceae | Field eryngo | Whole |
| Forb_RO | *Chenopodium* | Chenopodiaceae | Goosefoot | Whole |
| Forb_RO | *Euphorbia glareosa* | Euphorbiaceae | Spurge | Whole |
| Forb_RU | *Genista tinctoria* | Fabaceae | Dyer's greenweed | Whole |
| Forb_RU | *Astragalus* sp. | Fabaceae | Milkvetch | Whole |
| Forb_RU | *Galium* sp. | Rubiaceae | Bedstraw | Whole |
| Forb_RU | *Cannabis ruderalis* | Cannabaceae | Hemp | Whole |
| Forb_RU | *Thymus* cf. *hirsutus* | Lamiaceae | Thyme | Whole |
| Shrubs | | | | |
| Shrub_RU | *Ephedra distachya* | Ephedraceae | Joint pine | Twigs |

ally compare the aspect ratio of various categories, we generated boxplots using the mean $L/W$ values.

To demonstrate the applicability of experimental charcoal morphologies and morphometrics for the identification of 5 fuel burnt in sedimentary records, we randomly selected five Holocene samples from a core taken from Mangalia Herghelie wetland, Romania (43.838056° N TS12, 28.583333° E), and six samples from wetland sites near the Karagaily-Ayat River (profile Tom 52.864648° N, 60.222420° E and profile 10 SH 52.858475° N, 60.226214° E), where our plant material samples were collected (Fig. 1). We first bleached the samples for 6 h and washed them using a 125 µm sieve. Pollen analysis was additionally carried out on these samples to examine the correlation between the predominant vegetation 15 types and burnt fuel, as indicated by charcoal morphometrics. The pollen taxa were categorised into wood, including the pollen percentages of tree taxa; forbs, representing the pollen percentages of flowering herbaceous plants; and graminoids, encompassing both terrestrial (Poaceae, cereals) and wetland 20 (Cyperaceae, *Typha*) types.

## 2.2 Numerical analysis

The medians and standard deviations of the following charcoal morphometrics ($L/W$, $W/L$, $A/P$) were aggregated for each species, growth habit and plant type for all burn temperatures and displayed as boxplots. We used a two-tailed 25 Mann–Whitney test to determine whether the medians of the charcoal morphometrics of the growth groups and plant types were equal (Table S1). This test does not assume a normal distribution, only similar distributions in both groups.

## 3 Results                                                                 30

### 3.1 Fuel-dependent influence of temperature on charred-mass production

All plant material burnt had a typical charcoal appearance, i.e. black, opaque, angular, planar (Whitlock and Larsen, 2002), after being subjected to a temperature of 250 °C. 35 Most materials turned to ash at 400 °C, and all plant tissue became ash at 450 °C. Plant material burnt at higher temperatures (400–450 °C) tended to break more easily dur-

ing manipulation. Grouped by growth form, the percentage of charred mass retained at 300 °C (intermediate temperature) was lower for graminoids (34 %) than for shrubs (39 %) and forbs (42 %) (Fig. 2b; Table S2). Within the forb group, members of Asteraceae (*Anthemis, Xeranthemum, Achillea, Artemisia*) and *Chenopodium* had greater charred mass retained than other forb families (Lamiaceae, Fabaceae). Grouped by plant parts, the percentage of charred mass retained at 300 °C was lower for grass stems and leaves (33 %) than grass inflorescence (37 %; Fig. 2b; Table S2). This trend in the loss of charred mass was consistent at all temperatures (Fig. 2af TS13; Table S2).

## 3.2 Fuel-dependent variations in aspect ($L/W$; $W/L$) and area-to-perimeter ($A/P$) ratios

Grouped by growth form, the aspect ratios ($L/W$) of grass charcoals at all temperatures combined were consistently more elongated (mean $= 4.0 \pm 2.5$, median $= 3.1 \pm 1.4$) than for forbs (mean $= 3.1 \pm 2.2$, median $= 2.4 \pm 1.2$) and twigs of shrubs (mean $= 3.5 \pm 2.3$, median $= 3.1 \pm 2.3$; Table S2). These trends in $L/W$ are valid for all temperature increments (Fig. 3; Table S2). Within forbs, Asteraceae has more narrow leaves and shows high values in the $L/W$ ratio relative to other forb families, i.e. Fabaceae and Lamiaceae (Fig. 3). When grouped by plant parts, the $L/W$ ratio of grass stems and leaves (mean $= 4.2 \pm 2.8$, median $= 3.4 \pm 2.8$) at all temperatures combined was higher than that of grass inflorescence (mean $= 2.9 \pm 1.9$, median $= 2.6 \pm 1.9$; Fig. 3). Although individual measurements varied greatly, the mean values among species of grass and forbs are consistent (File S1 in the Supplement TS15; Table S2). The Mann–Whitney test confirmed that the $L/W$ ratios of grass were significantly different from forbs and that grass stems and leaves were significantly different from those of grass inflorescence at all temperatures (Table S1). The width-to-length ratio ($W/L$), an inverse way to report the aspect ratio of charred particles in the literature, shows that grass charcoal is narrower (mean $= 0.38 \pm 0.2$, median $= 0.3 \pm 0.2$) than forbs (mean $= 0.42 \pm 0.2$, median $= 0.4 \pm 0.2$), and grass stems and leaves (mean $= 0.34 \pm 0.24$, median $= 0.29 \pm 0.24$) are narrower than graminoid inflorescence (mean $= 0.42 \pm 0.16$, median $= 0.40 \pm 0.16$) and twigs of shrub (mean $= 0.36 \pm 0.18$, median $= 0.32 \pm 0.18$; File S2; Table S2).

The area-to-perimeter ratio ($A/P$), which describes more regular (smaller $A/P$ values) versus irregular (larger $A/P$ values) surfaces of charcoal, was only slightly smaller for grass (mean $= 50 \pm 30$, median $= 45 \pm 30$), stems and leaves (mean $= 47 \pm 29$, median $= 40 \pm 29$), and inflorescence (mean $= 52 \pm 25$, median $= 40 \pm 25$) than forbs (mean $= 53 \pm 25$, median $= 43 \pm 34$) and twigs of shrubs (mean $= 54 \pm 28$, median $= 44 \pm 35$; File S3; Table S2). There is also a clear tendency for $A/P$ ratio to decrease with increased temperature.

## 3.3 Finer diagnostic features of the charcoal morphologies of various fuel types

Graminoid charred particles from stems and leaves were consistently flat, rectangular and elongated (Fig. 4a). They mostly broke parallel to the long axis, resulting in elongated pieces with straight margins. They can also appear as featureless, long and thin filaments. The most common preserved surface features were rectangular epidermal cells or contained oval voids, reticulated or mesh patterns, and/or isolated veins.

Charcoal particles from the forb stems were cylindrical and rectangular, whereas fragments of leaves were mostly polygonal. Forbs with pinnate leaf shapes (Asteraceae, *Eryngium*) were generally more elongated than members of the Fabaceae and Lamiaceae families, which had more rounded and polygonal leaf shapes (Fig. 4b). Edges were undulate, smooth or denticulate. Surface textures were generally smooth (featureless); however, charcoal burnt at higher temperatures and those from sedimentary leaf charcoal had more frequently visible venation and ridges. When broken, these particles showed voids and reticulated mesh patterns. Twigs of *Ephedra* shrub showed both quadrilateral shape and layered or foliate structure. The edges and surface textures were smooth (Fig. 4b).

## 3.4 Morphometrics and finer diagnostic features of fossil charcoal

The aspect ratio of the sedimentary charcoal samples from Mangalia Herghelie, Romania, varied between 3.0 and 5.5 ($L/W$) or 0.35–0.43 ($W/L$), whereas $A/P$ ratio was between 28 to 40 (Table 2). Samples with elongated ($L/W$) and narrower particles ($W/L$) contained abundant morphologies of graminoid leaves (34 %–44 %) and graminoid and forb stems (44 %–50 %) as well as pollen from graminoids and forbs but had no charred wood morphologies (Table 2). The Tom and SH sedimentary charcoal profiles from Russia showed aspect ratios ranging between 3.6–5.0 ($L/W$) and 0.3–0.4 ($W/L$) and $A/P$ ratios between 23 to 34 and numerous particles larger than 1 mm (Table 2). Samples with the highest $L/W$ ratios showed the highest proportion of graminoid leaves (41 %–55 %) and high proportions of graminoid and forb stems (37 %–50 %) morphologies. There was a low proportion of wood charcoal morphologies (6 %–12 %) but a highly variable tree pollen percentage (16 %–41 %).

## 4 Discussion

### 4.1 The influence of combustion temperature on charcoal production for graminoids and forbs

The relationship between charcoal production and the quantity of unburned biomass and fire temperature varies among

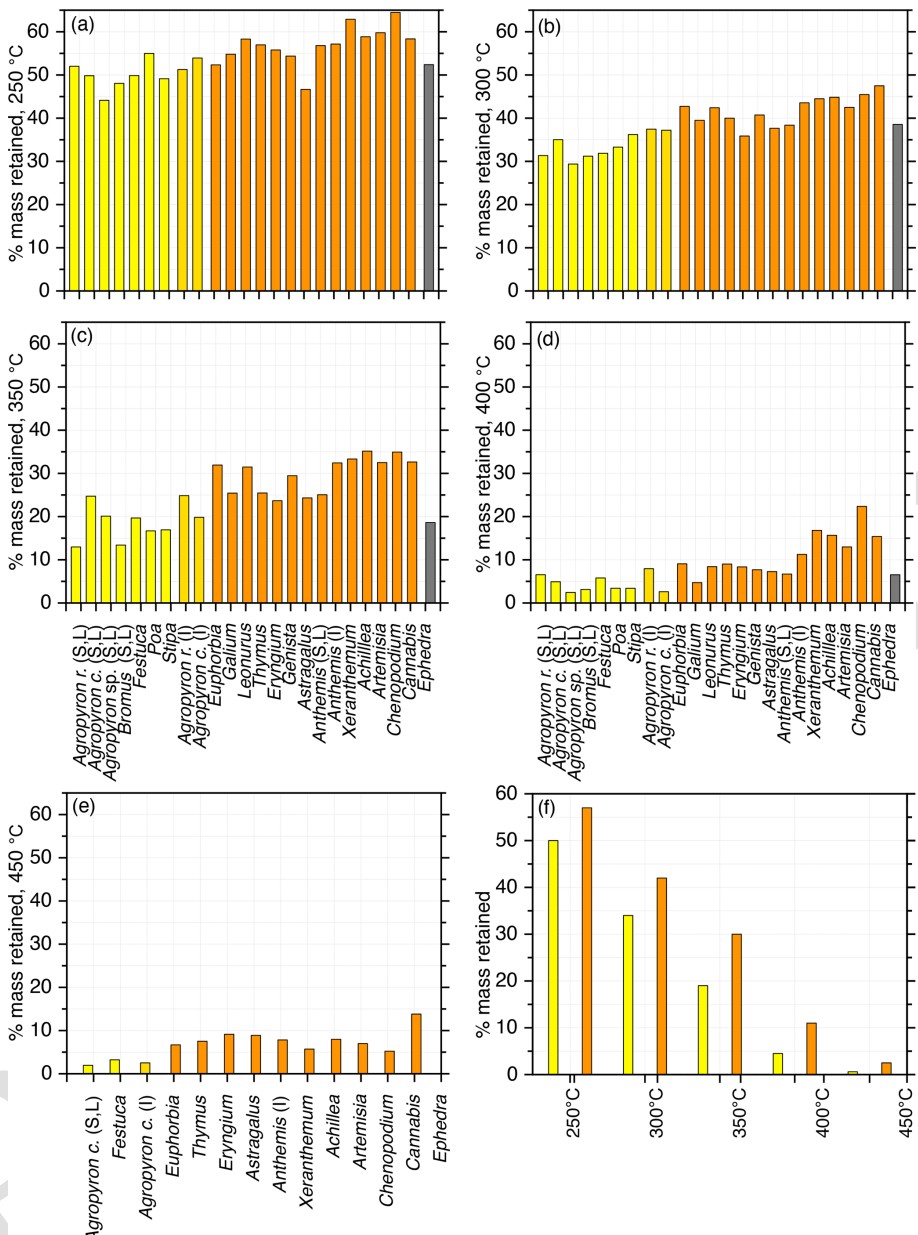

**Figure 2.** The percent of charred mass retained after burning individual plant taxa in a muffle oven at 250, 300, 350, 400, and 450 °C (**a–e**). The abbreviations used in the figure are S & L – stem and leaf, I – inflorescence; where this is not noted it implies mixed plant parts. Mean charred mass values grouped by grasses and forbs, the main growth habits, at increasing temperature from 250 to 450 °C are also presented (**f**). Grass taxa are represented in yellow (light yellow for S & L, dark yellow for I), forbs in orange, and shrubs in grey. The full names of plant species burnt and their geographic origins are presented in Table 1.TS14

fuel types due to variations in biomass moisture content, density, structure, chemistry and leaf surface-area-to-volume ratio (Walsh and Li, 1994; Hudpith et al., 2018TS17; Simpson et al., 2016). These are considered essential plant traits that influence the key components of fuel flammability, namely ignitability, combustibility and fire spread (Simpson et al., 2016; Pausas et al., 2017). While laboratory burning experiments only approximate some aspects of the heating conditions of vegetation, they provide valuable information on the amount of charcoal production with incremental rises in the fire temperature. Therefore, understanding species-specific charred mass production is crucial for interpreting charcoal-based fire reconstructions.

Our experimental charcoal production shows that grasses, particularly their leaves and stems, produced lower amounts of charcoal per unit biomass than grass inflorescences, forbs

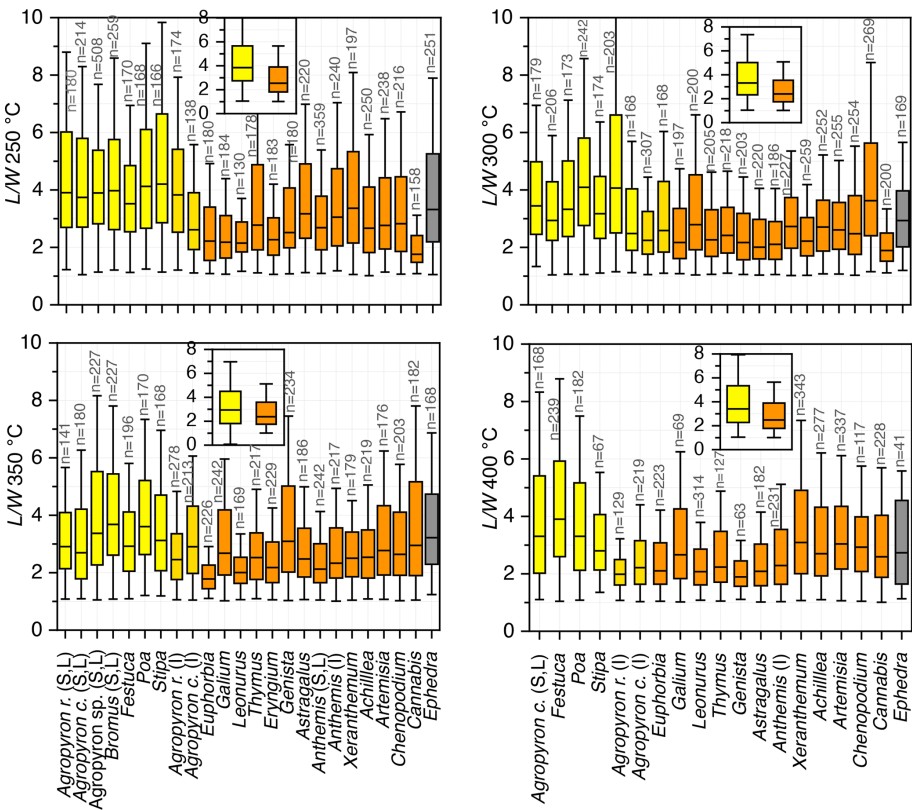

**Figure 3.** The median aspect ratios ($L/W$) of charred particles from the individual taxa combusted at 250, 300, 350 and 400 °C, respectively. Insets in each panel represent the median values of grass and forb taxa, with the same colour scheme as in individual measurements. Boxplots represent the distribution of data as follows: the horizontal line in each box denotes the median, the upper quartile is the median value of the upper half of the data points, the lower quartile is the median value of the lower half of the data points, and whiskers represent the respective minimum and the maximum values. Individual and mean $L/W$ values for each temperature are presented in Table S2, whereas mean $L/W$ values for all temperatures combined are shown in Fig. 5. Abbreviations and colour coding as in Fig. 2. $n$ indicates the number of charcoal particles measured for each taxon and temperature.

and shrub wood (Fig. 2; Table S2). Grasses also lost their charred mass more rapidly with increasing burning temperature, from 45 %–55 % at 250 °C to 1 %–3 % at 450 °C compared to forbs, 45 %–65 % at 250 °C to 5 %–15 % at 450 °C. Among forbs, members of the Asteraceae family and *Chenopodium* have higher charcoal production than members of the families Fabaceae and Lamiaceae. These results support laboratory burning experiments in a muffle oven and under open-flame conditions on several grass species from forest–steppe in North America showing that the charcoal mass retention of grass was significantly lower than that of wood (Umbanhowar and McGrath, 1998), a pattern also replicated in a calorimetric combustion study including graminoid, wood, needles and deciduous leaves (Hudspith et al., 2018). Similarly, burning experiments in a muffle oven on wetland graminoids from Alaska tundra (Pereboom et al., 2020) and Siberian taiga (Feurdean, 2021) demonstrate that these graminoids have significantly lower charred-mass retention per unit biomass than other growth forms (forbs, shrubs/trees) or plant parts (leaves and wood). Another ex-

perimental study conducted in laboratory settings using undried grasses has shown that the fuel moisture content significantly impacts ignition probability: the higher the moisture content, the longer the ignition times. Biomass quantity predominantly governs combustibility and fire spread, whereas biomass density exerts a less pronounced effect on flammability (Simpson et al., 2016). In our experiments, utilising considerably smaller quantities of lightly dried plant material (40 °C), the lower bulk density and water content of graminoid compared to forb biomass may explain the observed reduction in grass charcoal production, aligning with findings from similar studies. Incorporating a broader range of fuel moisture conditions ~~that closely resemble real-world scenarios~~ may need to be considered to enhance the relevance of experimental laboratory research on charcoal production. This aspect may be particularly relevant when distinguishing between fires in terrestrial and wetland habitats dominated by graminoids. Given their high fuel moisture, wetland graminoids may exhibit higher charcoal production than dry

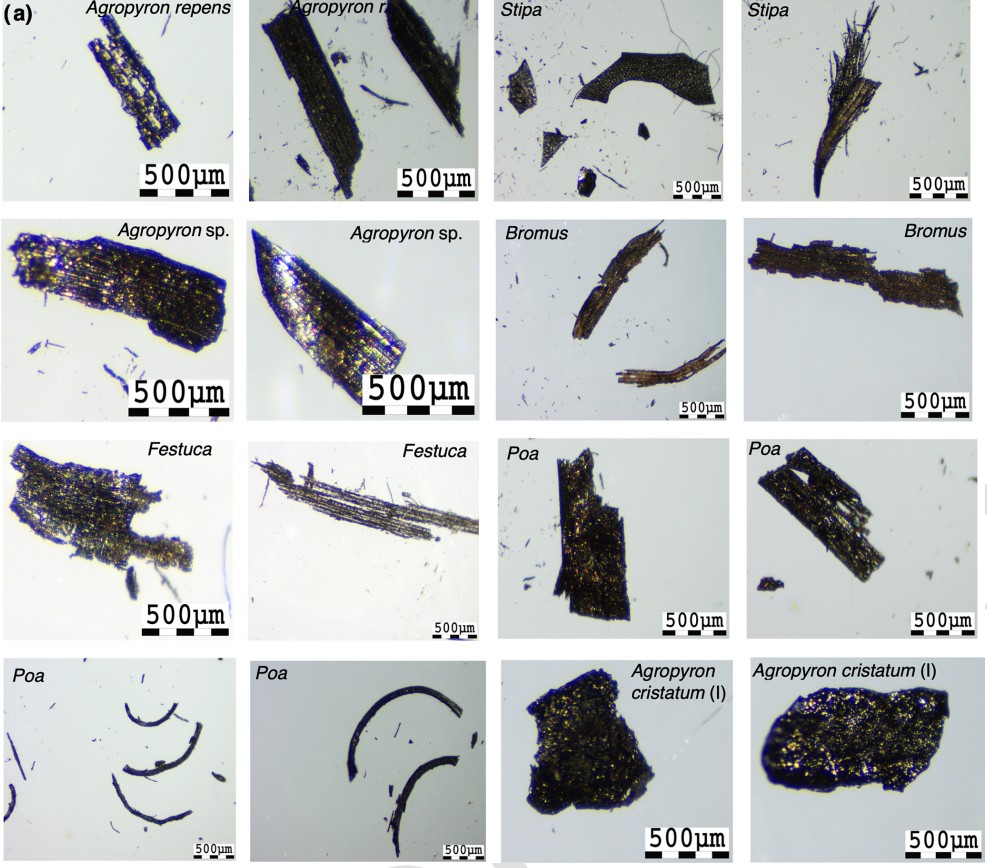

**Figure 4.**

graminoids, which may be overlooked when using dried samples.

The chemical composition of plants, including their mineral constituents, lignin, cellulose, nitrogen and phosphorous content, plays a pivotal role in charcoal production. In laboratory experiments, forbs yielded more charcoal than wood, likely because of their cellulose- and hemicellulose-rich composition pyrolysing at lower and narrower temperature ranges (200 to 400 °C), as opposed to wood, which is rich in lignin (160 to 900 °C; Yang et al., 2007). However, in the oven settings, plant tissue is converted to ash more rapidly (typically between 450 and 500 °C) compared to natural fire conditions (around 800 °C) due to reduced influences of flame dynamics and turbulent airflow (Belcher et al., 2015; Hudpith et al., 2018). One possible explanation for the lower wood charcoal production may be attributed to the faster combustion rates observed in oven settings, resulting in wood being ashed at the lower end of its temperature range. Field research has provided evidence that prolonged combustion through smouldering can also transform wood into white ash (Courtney Mustaphi et al., 2022). Investigations involving chemical analyses of unburnt biomass can enhance our understanding of how plants' chemical composition influences charcoal production.

## 4.2  Fuel-dependent variability in charcoal morphometrics

We observed consistent $L/W$ (elongation) or $W/L$ (narrowness) ratios of charred fragments among our Eurasian species of the same plant group (genus or family), growth type and plant parts at all temperatures (Fig. 3, Table S2). This consistency suggests that aspect ratio could be a useful morphometric for fuel type separation in palaeoenvironments dominated by grasslands. We found that the charred Eurasian grasses are more elongated ($L/W$ $4.0 \pm 2.5$) and narrower ($W/L$ $0.38 \pm 0.2$) than forbs ($L/W$ $3.1 \pm 2.2$, $W/L$ $0.42 \pm 0.2$), in agreement with a global compilation for graminoids ($5.4 \pm 2.3$; grass exclusively $4.3 \pm 1.7$) and forbs ($2.9 \pm 0.4$; Fig. 5; Table 3). We also found differentiation of aspect ratio between parts of the same plant, with grass stems and leaves being more elongated ($4.2 \pm 2.8$) and narrower ($0.34 \pm 0.24$) than grass inflorescence ($2.9 \pm 1.9$; $0.42 \pm 0.1$; Fig. 3; Tables S1, 2). Thinner-leaved forb species in the Asteraceae family also have a higher aspect ratio than members of the families Fabaceae and Lamiaceae, which have more polygonal or rounded leaves. Our $L/W$ values for temperate grasses agree with individual experimental laboratory burning of *Poa trivialis* (3.7) from a UK botanical

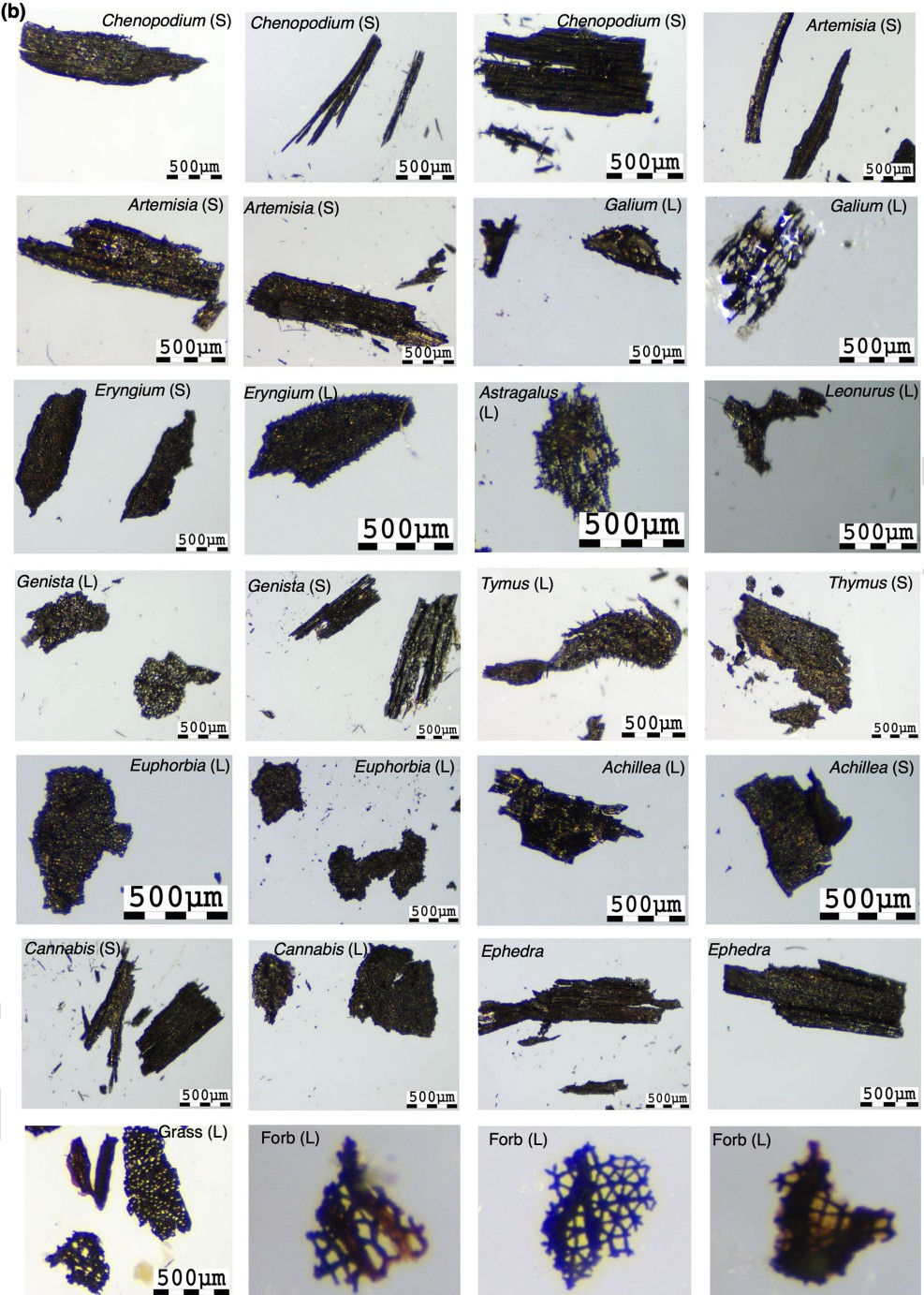

**Figure 4.** Photomicrographs of characteristic charcoal morphotypes under stereomicroscope (4×). **(a)** Grass and **(b)** forbs. The last raw CE4 in **(b)** presents sedimentary charcoal. L – leaf, S – stem, I – inflorescence.

garden (Crawford and Belcher, 2014) and several temperate grass species from North America (3.6 from furnace versus 4.8 from open-flame combustion; Umbanhowar and McGrath, 1998). Crucially, our global compilation shows that grasses, *Poa, Stipa, Agropyron*, etc., are shorter and bulkier ($4.3 \pm 1.7$) than wetland graminoids, Cyperaceae, *Eriopho-rum* and *Phragmites* ($7.6 \pm 0.6$ for boreal and $6.4 \pm 3$ global wetland graminoids; Fig. 5, Table 3). The consistency in $L/W$ values for graminoids from temperate grasslands, on the one hand, and of wetland graminoids, on the other hand, may suggest differences in the evolutionary developments in the physiology of dryland graminoids relative to that of

**Table 2.** Charcoal morphometrics: the mean aspect ratio ($L/W$, $W/L$) and area-to-perimeter ($A/P$) ratio and percentage of main charcoal morphotypes in the Holocene samples from Mangalia Herghelie, Black Sea, SE Romania, and Konoplyanka, sites Tom and SH, Trans-Urals, Russia. TS16

| Age (cal yr BP) | 2350 | 2950 | 3150 | 3950 | 5850 | |
|---|---|---|---|---|---|---|
| Mangalia Herghelie | | | | | | |
| Charcoal morphometrics | | | | | | |
| $L/W$ | 5.5 | 4.4 | 3.2 | 3.3 | 3.0 | |
| $W/L$ | 0.37 | 0.34 | 0.43 | 0.41 | 0.35 | |
| $A/P$ | 40 | 34 | 37 | 37 | 28 | |
| Charcoal morphologies (%) | | | | | | |
| Wood | 0 | 0 | 7 | 6 | 4 | |
| Leaf (forb & tree) | 7 | 22 | 12 | 15 | 7 | |
| Leaf graminoid | 43 | 34 | 26 | 34 | 48 | |
| Stem graminoid & forb | 50 | 44 | 55 | 45 | 41 | |
| Pollen (%) | | | | | | |
| Tree | 26 | 26 | 29 | 34 | 36 | |
| Shrub | 0 | 0.5 | 1 | 1 | 0.6 | |
| Grass | 23 | 25 | 38 | 7 | 23 | |
| Graminoid wetland | 23 | 9 | 11 | 2 | 25 | |
| Forb | 52 | 48 | 32 | 58 | 40 | |
| Age (cal yr BP) | 1700 | 1800 | 1750 | 2050 | 2830 | 4140 |
| Konoplyanka | Tom | Tom | Tom | SH | SH | SH |
| Charcoal morphometrics | | | | | | |
| $L/W$ | 5.0 | 3.7 | 3.6 | 4.0 | 4.8 | 4.1 |
| $W/L$ | 0.3 | 0.4 | 0.36 | 0.3 | 0.3 | 0.34 |
| $A/P$ | 24 | 34 | 31 | 27 | 23 | 29 |
| Charcoal morphologies (%) | | | | | | |
| Wood | 6 | 10 | 10 | 12 | 7 | 8 |
| Leaf (forb & tree) | 2 | 0 | 0 | 1 | 1 | 2 |
| Leaf graminoid | 55 | 17 | 17 | 23 | 41 | 29 |
| Stem graminoid & forb | 37 | 73 | 73 | 64 | 50 | 61 |
| Pollen (%) | | | | | | |
| Trees | 20 | 41 | 16 | 31 | 36 | 19 |
| Shrub | 0 | 0 | 0 | 0 | 0 | 0 |
| Grass | 31 | 24 | 50 | 16 | 19 | 23 |
| Graminoid wetlands | 7 | 3 | 2 | 9 | 1 | 2 |
| Forb | 49 | 35 | 34 | 53 | 45 | 58 |

wetlands graminoids (Hedges and Mann, 1979; Vachula et al., 2021). Both charcoal types have elongated vascular bundles connected to the occurrence of veins parallel to the long axis (Umbanhowar and McGrath, 1998). However, the stems of grasses are more robust, with developed resistance tissue, than those of wetland graminoids. Wetland graminoid mass loss during senescence may also explain their lesser robustness (Tanner, 2001; Vernescu et al., 2005). The comparison of $L/W$ ratio of charcoal particles produced in a muffle oven with those generated under open-flame conditions reported in the literature reveals slightly longer $L/W$ values for particles produced in the open flames. Although this observation underscores the varying fragility of charcoal produced under different burning conditions, it also highlights that the relative proportion of the $L/W$ ratio between different fuel types remains consistent.

In contrast to Lestienne et al. (2020), working in Mediterranean ecosystems, we found little differentiation in $A/P$ between taxa and plant parts. The $A/P$ for grass charcoal ($50 \pm 25$) shows that they only have a slightly more regular form than forbs ($52 \pm 30$), suggesting that $A/P$ may not be as helpful for fuel differentiation in ecosystems dominated by grasses and forbs (Tables S2 and S3 TS19). However, there is a tendency for $A/P$ ratio to decrease with increasing temperature, which is more evident for forbs (from 59 to 40) than for grass (from 53 to 44). This characteristic, combined

**Table 3.** Comparison of $L/W$ values of experimental charcoal produced from known plants in this study with those compiled from the literature by Vachula et al. (2021) and new Feurdean (2021) results. Graminoid all global includes grass and graminoid wetland type. $L/W$ value for non-woody type combines charcoal of grasses and forbs, whereas the $L/W$ value in parentheses combines grasses, wetland graminoids and forbs. The wood sums the $L/W$ values of trunks and twigs.

| Fuel type | Zone | $L/W$ | References |
|---|---|---|---|
| Graminoid (grass) | Eurasia | $4.0 \pm 2.5$ | This study |
| Graminoid (grass) | Global | $4.27 \pm 1.7$ | This study, Vachula et al. (2021) |
| Graminoid (wetland) | Boreal | $7.64 \pm 0.6$ | Pereboom et al. (2020), Feurdean (2021) |
| Graminoid (wetland) | Global | $6.4 \pm 3$ | Vachula et al. (2021) |
| Graminoid (all) | Global | $5.4 \pm 2.3$ | See above |
| Forb | Boreal | $3.1 \pm 2.2$ | This study |
| Forb | Global | $2.9 \pm 0.4$ | This study, Vachula et al. (2021), Feurdean (2021) |
| Non-woody (graminoid & forb) | Eurasia | $3.6 \pm 2.4$ | This study |
| Non-woody (graminoid & forb) | Global | $3.6 \pm 1.1$ $(4.3 \pm 1.6)$ | This study, Vachula et al. (2021), Feurdean (2021) TS18 |
| Leaf (forbs, shrub & trees) | Global | $2.1 \pm 0.1$ | Vachula et al. (2021), Feurdean (2021) |
| Wood (shrub & tree) | Global | $3.4 \pm 0.4$ | This study, Vachula et al. (2021), Feurdean (2021) |

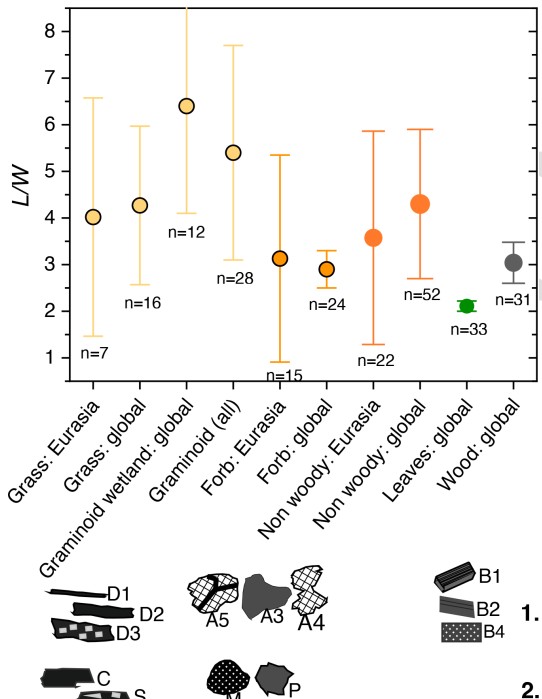

**Figure 5.** Mean $L/W$ values from experimental charcoal produced from known plants for main growth habits (terrestrial grasses, wetland graminoids, forbs) and plant types (leaves, wood) from this study (Eurasia) and the compilation from the literature (see details Table 3). Comparison with selected charcoal morphotypes of (1) Courtney Mustaphi and Pisaric (2014) and (2) Enache and Cumming (2006) is also shown.

with the fact that fossil samples have much lower $A/P$ values (23–40) than those from experimental burns, suggests higher wildfire temperatures than experimental burns and/or some edge rounding (degree angles are eroded) during transportation and deposition. A decreased surface area and increased roundness with transportation have also been documented (Vannière et al., 2003; Crawford and Belcher, 2014; Courtney Mustaphi et al., 2015).

### 4.3 Finer diagnostic features of the charcoal morphologies for fuel-type identification comparison with keys from the literature

Feurdean (2021) provided detailed examinations of the morphological characteristics of experimentally produced charcoal from various taxa and plant parts from the boreal zone, including wetland graminoids, leaves of mosses, ferns, forbs and trees, and the wood of trees and shrubs. The present study improves the link between the grass and forb species from Eurasian open ecosystems and their charcoal morphologies. Similar to charred wetland graminoids, charred temperate graminoids can be distinguished by their flat appearance and tendency to break into thin filaments (Fig. 4a). The grass stems and leaves were not burned separately in this study. However, comparison with previous publications and atlases (Schweingruber, 1978 TS20) indicates that grass leaves can often be distinguished by rectangular epidermal cells, reticulate meshes and oval voids of former epidermal stomata, whereas grass stems display layered, foliated textures that share the appearance of wood (Fig. 4a). This feature contrasts with wetland graminoid charcoal, which has a more elongated, fragile form (Feurdean, 2021). The grass inflorescence charcoal is irregular in shape with foliated polygons (Fig. 4a). The grass charcoals produced here are like type B (B2, B3, B4, B5), which is attributed to wood, type D (D1, D2, D3, F),

which is attributed to monocotyledon leaves, and type E (E1, E2) mostly related to seeds (Fig. 5; Enache and Cumming, 2006; Courtney Mustaphi and Pisaric, 2014).

Typical features of forb leaves include their polygonal shapes (Fig. 4b). Netted venation with three branches diverging from a node and surfaces characterised by void spaces are visible in particles charred at higher temperatures as well as sedimentary particles. Forb stems are cylindrical (unbroken) and layered with foliated textures. Our forb charcoals are most like morphology type A (A1, A2, A3, A4, A5), which is mostly attributed to leaves and wood, type B (B2, B3, B4) and type C (C5, C6), which are connected to leaves, stems and veins (Courtney Mustaphi and Pisaric, 2014), whereas some fragments resemble types M and P (Enache and Cumming, 2006).

In summary, our charcoal morphological analysis demonstrates that grass leaves are easily distinguished from forb leaves. However, there is an overlap between stems of grass and forb, indicating that the two groups are difficult to discriminate based on charcoal morphologies. There is also some overlap between stems of grass and forb with wood morphologies, particularly in more mature plants, raising the possibility of misclassifying some stems of herbs and grass as wood morphologies (Fig. 4). To improve the accuracy of attributing charcoal morphologies to specific fuel types, we recommend enhancing the classification of charcoal morphologies by further examining particles generated from burning known plant materials.

## 4.4 The morphometrics and morphologies of fossil charcoal particles: implications for fuel-type identification

All analysed sedimentary charcoal samples show a significant presence of graminoid and forb morphologies, with aspect ratios ranging between 3.0 and 5.5, though the aspect ratio is slightly higher at the Russian sites (Table 2). These findings align well with the pollen records at these sites, which also demonstrate a high proportion of these plant types (Table 2). Nonetheless, notable discrepancies between the Romanian and Russian sites are also evident. At the Mangalia Herghelie site in Romania, samples with the highest aspect ratios (4.4–5.5) exhibit a considerable abundance of graminoid leaf morphologies (34 %–44 %) and pollen from both dry (23 %–25 %, Poaceae, cereals) and wetland (9 %–23 %, Cyperaceae, *Typha*) graminoids. Additionally, the stratigraphy at this site indicates the presence of peat with abundant remains of wetland graminoid *Phragmites*, which belongs to the Poaceae family, and its pollen morphology is, therefore, Poaceae-type. Based on the evidence above, we propose that fires likely occurred in both dry grassland areas and within the reed and sedge vegetation that developed around and on this site. Historically, burning reed vegetation to access water was a common practice among past societies (Feurdean et al., 2013; Sim et al., 2023). The

low percentages of wood morphologies (0 %–7 %) and low to moderate tree pollen percentages (26 %–36 %) in samples from Mangalia Herghelie indicate the scarcity of local woody vegetation, and the burning of woody fuel was limited. In contrast, the sites in Russia display a slightly higher percentage of woody morphologies (6 %–12 %) that do not always match the highly variable abundance of tree pollen percentages (19 %–41 %). The presence of fortified and unfortified settlements, along with kurgans dating back to the Middle Bronze Age, has been documented in the vicinity of these sites. A nomadic lifestyle is also postulated for the Iron Age in this region (Stobbe et al., 2016). It is, therefore, plausible that household burning, combined with the selective water transport of charcoal, has increased representation of woody charcoal in the samples from this region.

Our findings indicate that the charcoal morphologies and morphometrics largely reflect the dominant plant communities shown by the pollen record, particularly for herbaceous vegetation suggesting a robust link between the dominant vegetation type and fuel burnt in grass-dominated environments. However, the proportion of tree pollen matches less than that of woody charcoal, which suggests atmospheric transport of tree pollen over longer distances compared to woody charcoal. These observations also highlight the complex interplay between local vegetation and charcoal composition with human activities and the need to consider site-specific factors when interpreting charcoal records.

## 4.5 Application of charcoal morphologies and morphometric for fuel- and fire-type reconstructions in grassland-dominated ecosystems

Our experimental production of charcoal has three primary applications for past fire reconstruction in grass-dominated systems. Firstly, it can improve the identification of fuel types based on charcoal morphologies and morphometrics. Secondly, it can determine the effect of biomass and fire temperatures on taxa-specific charcoal production. Thirdly, it can assess the influence of charcoal shape from different fuel types on the spatial scale of fire regime reconstructions.

Our analyses demonstrate that charcoal morphologies provide a finer distinction between fuel types than morphometrics. Grass leaf morphologies can be easily distinguished from forb leaves. However, morphologies of stems of grass and forb are more challenging to differentiate from each other. Charcoal morphometry also shows promising results in establishing threshold values for discriminating individual fuel types. Limitations, however, may arise when analysing charcoal assemblages from mixed fuel sources due to overlapping $L/W$ values. We identified a mean $L/W$ value of $4.0 \pm 2.5$ for laboratory produced grasses and $3.1 \pm 2.2$ for forbs charcoal (Fig. 5; Table 3). These values largely agree with the values generated from a literature compilation of laboratory-produced charcoal, $5.4 \pm 2.3$ for graminoids ($4.3 \pm 1.7$ for grasses; $6.4 \pm 3$ for wetland graminoids),

$2.9 \pm 0.4$ for forbs and $2.2 \pm 0.1$ for leaves (Fig. 5; Table 3). While our study used only a single woody shrub species (*Ephedra*; $L/W$ $3.5 \pm 2.3$), the literature compilation of woody particles revealed $L/W$ $3.04 \pm 0.4$, meaning an overlap with non-woody vegetation, especially forbs (Table 3; Fig. 5). We propose that $L/W$ values above 3 ($W/L$ below 0.40) can indicate predominantly charcoal morphologies from grass and forbs in temperate grassy-dominated ecosystems. However, values can be higher in grass relative to forb-dominated ~~vegetation within this ecosystem~~. Our proposed cut-off values to separate non-woody from woody fuels, based on experimental studies with known fuel types, are closest to those applied to a wide range of ecosystems in Europe (Marriner et al., 2019; Lebreton et al., 2019) and Canada (Umbanhowar and McGrath, 1998), and partially to those from the USA (Leys et al., 2017) and Asia (Miao et al., 2019, 2020), and differ the most from those commonly applied in Africa (Aleman et al., 2013; Daniau et al., 2007, 2013, 2023; Thevenon et al., 2003), particularly those using $L/W$ above 2 or $W/L > 0.5$ for non-woody vegetation (see also review by Vachula et al., 2021). Therefore, the morphometric cut-off values are not universally valid but vary with the dominant ecosystem type. Hydroclimatic conditions and taphonomy have also been identified to contribute to these differences (Courtney Mustaphi and Pisaric, 2014). Consequently, the analysis of charcoal morphotypes and morphotype assemblages in sedimentary records requires the adoption of a more adaptable framework.

Comparing sedimentary charcoal morphometrics and morphologies with pollen and plant macrofossils can aid in interpreting the fuel source~~, and our assessment in the Eurasian steppe sites shows great potential~~ (Table 2). For example, comparison of the abundance of graminoid morphologies with that of pollen of grasses and wetland graminoids improved the determination of the origin of the charred graminoids. Vice versa, variability in charred morphologies and morphometrics can signal changes in the main vegetation composition over time. The morphologies of graminoids and forbs alongside high aspect ratios at our fossil sites appear to indicate the expansion of ~~open~~ grassy habitats. Conversely, the expansion of woody vegetation may be reflected by an increase in woody morphologies and a decrease in particle aspect ratio; however this relationship is not always straightforward.

Secondly, the quantity of charcoal produced varies depending on the traits of the vegetation burned and fire temperature, which can induce biases in charcoal-based fire reconstructions. Our experimental burnings using lightly dried biomass reveal that grass fuels from all taxonomic groups exhibit low post-fire charred-mass retention and rapidly lose mass with increasing temperature. Grasses have developed strategies for rapid flammability, including quick curing (dry out), easy ignition, rapid burning and fire spread (Pausas et al., 2017; Bond et al., 2019[TS21]; Stevens et al., 2022[TS22]). Given that grasses tend to dry out rapidly after flowering,

they are likely to leave little charcoal or turn into ashes even in relatively low-intensity field fires, leading to an underrepresentation of grass charcoal in sedimentary records. On the other hand, wetland graminoids such as reeds and sedges, thriving in areas with elevated soil and fuel moisture, may possess higher resistance to combustion compared to terrestrial grasses, thus contributing a higher proportion to locally produced charcoal. In contrast to grasses, forbs tend to retain a higher proportion of their biomass in our experiments even when subject to high temperatures, suggesting that the forb component of grasslands is more likely to be incorporated into sediments and may also lower the $L/W$ values. Consequently, lower quantities of charred graminoids relative to forbs and wood may signify high-intensity fires occurring within a tree–grass matrix rather than an absence of grass fires. These complexities underscore the importance of considering the diverse behaviours of different grassland components when studying their flammability and response to fire.

Thirdly, dispersal models accounting for shapes, sizes and densities of charcoal show that small differences in shape and density of charcoal can cause substantial differences in particle source area (Vachula and Richter, 2018; Vachula and Rehn, 2023). Elongated, lighter particles such as those from graminoids tend to have higher residence time in the atmosphere and can travel longer distances than spherical or heavier particles, such as those from leaves or wood (Clark and Hussey, 1988; Pisaric, 2002[TS23]; Vachula and Richter, 2018; Courtney Mustaphi et al., 2022[TS24]; Vachula and Rehn, 2023). Charcoal particles produced by grassland fires tend to be smaller than those produced by forest fires (Belcher et al., 2005; Leys et al., 2017), and our results show that the size of charcoal particles decreased with the increased fire temperature. Consequently, sedimentary charcoal in grass-dominated and mixed woody–grass ecosystems may represent a more regional fire history compared to forested ecosystems. This in turn may affect the reconstruction of local fire events using the decomposition method in peak and background charcoal (Vachula and Rehn, 2023). The decomposition methods were originally developed for North American forest ecosystems characterised by mixed conifer trees (Gavin et al., 2005[TS25]; Higuera et al., 2009[TS26]). These forests typically experience high-intensity fires with fire return intervals exceeding 100 years. In contrast, grass fires are characterised by low intensity and high frequency, although our understanding of these metrics in temperate grasslands remains limited (Pausas et al., 2005[TS27]; Feudean et al., 2018[TS28]; Stevens et al., 2022[TS29]). A more precise determination of the form and density of charcoal from various fuel types can be invaluable for understanding its transport and interpreting the spatial scale of fire histories in the context of major ecosystem shifts.

## 5 Conclusions

This study provides novel insights into charcoal mass production, morphometrical aspects, and morphologies of several grass and forb species from the Eurasian grasslands. It highlights the practical application of morphometry and morphology in enhancing the understanding of fuel composition and fire dynamics in grass-dominated ecosystems.

The impact of fire temperature on charcoal production varies depending on the characteristics of the fuel type. Grasses exhibit significantly lower charcoal production and decrease more prominently with increasing fire temperature than forbs. This suggests an under-representation of grass morphologies in the sedimentary charcoal record relative to forbs. However, in field settings, wetland graminoids thriving in areas with elevated soil moisture may exhibit greater resistance to combustion and thus have a higher charcoal production. The decrease in the size of grass and forb charcoal particles with the increased fire temperature and the elongated form of grass charcoal suggests that reconstructing fire regimes in grass-dominated and mixed woody–grass ecosystems may provide insights into more regional fire history than forested systems.

Charcoal morphologies offer a more detailed taxonomical and plant part distinction for identifying the vegetation burnt (fuel ~~typos~~) types than morphometrics. However, both morphologies and morphometrics can complement each other in fuel identification. Leaf morphologies of grass can be easily distinguished from forb, but differentiating between the stem morphologies of grass and forbs presents challenges. Morphometric analysis indicates that graminoid charcoal particles are more elongated ($4.0 \pm 2.5$) than forbs ($3.1 \pm 2.2$); however, literature compilation shows some overlap with the global aspect ratio of wood ($3.04 \pm 0.4$). Nevertheless, $L/W$ values above 3 ($W/L$ below 0.40) may indicate predominantly herbaceous morphologies in temperate open ecosystems (steppe, forest steppe). In addition, grasses exhibit shorter aspect ratios ($4.3 \pm 1.7$) than wetland graminoids ($6.4 \pm 3$). The cut-off values may need adjustment and should not be universally applied to other ecosystems without a reasonable rationale. In research conducted within grass-dominated and mixed woody–grass systems, adopting lower threshold values for aspect ratios might be more appropriate than in ecosystems primarily dominated by wetland vegetation. As morphologies provide a more effective way of distinguishing charred particles from forbs, grasses and wood, we recommend integrating these analyses to enhance distinctions between groups with overlapping morphometrics.

Variability in charred morphology assemblages and aspect ratios also holds promise in reconstructing changes in vegetation composition, particularly in areas with poor pollen preservation. Our preliminary results suggest that the morphologies of graminoids and forbs are more closely linked to their local abundance as indicated by pollen. However, the relationship between woody charcoal and pollen may be more complex for trees, as their pollen can travel longer distances within the atmosphere compared to woody charcoal. A more precise determination of the form and density of charcoal from various plant types can be invaluable for understanding dispersal, transportation and fragility, with implications for the spatial scale of charcoal reconstructions.

A critical point arising from this study is the need to avoid assuming the universality of research findings and instead apply experimental approaches to characterising charcoal particles in new ecosystem types. Future efforts to determine fuel sources based on analyses of charcoal fragment morphologies and morphometries should consider the investigation of taxa and plant parts from tropical grasslands, which have been largely overlooked in previous studies. Future analyses that incorporate the use of new image recognition or artificial intelligence software to collect and identify charcoal particles could aid in the differentiation of fuel types at the particle scale. While analysis of sedimentary charcoal requires a degree of aggregation and/or human interpretation related to fuel type changes, automated approaches could allow for the identification of the fuel source of individual charcoal particles with greater accuracy.

Laboratory experiments that incorporate a broader range of plant traits, such as fuel moisture, chemical components (nitrogen, phosphorus and tannins) and plant architecture, pivotal in determining flammability, have the potential to contribute to a more comprehensive understanding of the variation in charcoal abundance and morphometrics from diverse fuel types. ~~Furthermore, it is essential to recognise that field conditions can substantially differ from controlled experimental settings, which may impact charcoal production, fragility and morphometry.~~ While it is challenging to replicate the complexity of real-world scenarios due to numerous uncontrollable factors, future research should consider incorporating field studies that account for factors such as fire spread ~~dynamics~~, variability fuel quantity and quality, and fluctuating temperature conditions within the experimental design.

*Data availability.* All raw measurements of length, width, area, perimeter and ratios of $L/W$, $W/L$ and $A/P$ for all taxa and temperatures are presented in File S1 in the Supplement. This dataset is stored at GFZ Data Services (https://doi.org/10.5880/fidgeo.2023. 035, Feurdean, 2023 TS30).

*Sample availability.* A limited amount of burnt plant material can be made available upon request.

*Supplement.* The supplement related to this article is available online: https://doi.org/10.5194/bg-20-1-2023-supplement.

*Author contributions.* AF designed the study and performed the burning experiments, the morphometrical, morphological and numerical analyses, and data presentation; DH, AS, MG and AF collected the modern plants and performed pollen analysis. RSV compiled the morphometrics database from the literature; AF wrote the manuscript with contributions from RSV; all the authors reviewed and edited the manuscript and agreed with its content.

*Competing interests.* The contact author has declared that none of the authors has any competing interests.

ther geographical representation in this paper. While Copernicus Publications makes every effort to include appropriate place names, the final responsibility lies with the authors.

*Acknowledgements.* We would like to thank Markus Rosensthil for help developing the code for automatic detection of charred particles and Doris Bergmann-Dörr for help with burning plant material in the muffle oven.

*Financial support.* This work was supported by the Deutsche Forschungsgemeinschaft (grant nos. FE_1096/6 and FE_1096/9). Richard S. Vachula was supported by start-up funds from Auburn University TS31 and Diana Hanganu by PN-III-P4-ID-PCE-2020-2282 (ECHOES TS32).

This open-access publication was funded by the Goethe University Frankfurt.

*Review statement.* This paper was edited by Petr Kuneš and reviewed by Abraham Dabengwa and one anonymous referee.

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

## Remarks from the language copy-editor

CE1  State was added.
CE2  Please note the slight spelling adjustment here.
CE3  Do you mean "Feather grass"?
CE4  Do you mean "row" (or perhaps column)?

## Remarks from the typesetter

TS1  Changed 2004 to 2005; please confirm.
TS2  Changed 2001 to 2002; please confirm.
TS3  Not mentioned in the reference list. Please check throughout the paper.
TS4  Do you mean Feurdean et al., 2021? Otherwise this reference is not mentioned in the reference list. Please check throughout.
TS5  Not mentioned in the reference list.
TS6  Not mentioned in the reference list. Please check throughout the paper.
TS7  Not mentioned in the reference list. Please check throughout the paper.
TS8  Please confirm addition of "et al." throughout.
TS9  Not mentioned in the reference list.
TS10  Not mentioned in the reference list.
TS11  Please confirm running title or provide an alternative.
TS12  Please confirm addition of degree symbol throughout.
TS13  Should it be "a, f"?
TS14  The composition of Figs. 2 and 4 has been adjusted to our standards.
TS15  Please check and confirm. Currently it is difficult to locate which table or figure can be found in which file of the supplement. Please consider renaming the files to make it clearer what can be found where (e.g. calling the Excel file, "File_S1" and the Word file with the tables and figures "File_S2-tables_and_figures"). If any changes are required to the text, please let me know.
TS16  Please confirm table layout. If any horizontal lines should be added or deleted, please let me know.
TS17  Not mentioned in the reference list. Please check throughout the paper.
TS18  Please confirm deletion of one instance of Feurdean (2021).
TS19  Please confirm.
TS20  Changed 1987 to 1978; please confirm.
TS21  Not mentioned in the reference list.
TS22  Not mentioned in the reference list.
TS23  Not mentioned in the reference list.
TS24  Not mentioned in the reference list.
TS25  Not mentioned in the reference list.
TS26  Not mentioned in the reference list.
TS27  Not mentioned in the reference list.
TS28  Not mentioned in the reference list.
TS29  Not mentioned in the reference list.
TS30  Please check DOI.
TS31  If available, please provide a grant number.
TS32  Does this grant refer to the European Commission, Horizon2020 framework programme, grant no. 727470?
TS33  Please ensure that any data sets and software codes used in this work are properly cited in the text and included in this reference list. Thereby, please keep our reference style in mind, including creators, titles, publisher/repository, persistent identifier, and publication year. Regarding the publisher/repository, please add "[data set]" or "[code]" to the entry (e.g. Zenodo [code]).
TS34  Not mentioned in the reference list.
TS35  Not mentioned in the text.
TS36  Not mentioned in the text.
TS37  Please check DOI.
TS38  Please check DOI.

TS39  Please provide a persistent identifier.
TS40  Please check DOI.
TS41  Not mentioned in the text.

Please note the remarks at the end of the manuscript.