# Peer review of "Charcoal morphologies and morphometrics of a Eurasian grass-dominated system for robust interpretation of past fuel and fire type"

_EGUsphere, 2023_

## Author Response (AR1)

*24 Sep 2023 **Associate editor decision: Reconsider after major revisions***
*by Petr Kuneš*
*Thank you for your elaborate comments on the reviews. Please proceed with major revision considering all comments.*

5.10.2023 Frankfurt am Main

Dear Editor-in-Chief and associate editor, Petr Kuneš

We would like to thank the two reviewers for their encouraging feedback and valuable comments, which have contributed to improving the current version of our paper: *Charcoal morphologies and morphometrics of a Eurasian grass-dominated system for robust interpretation of past fuel and fire type.* We have carefully considered and incorporated their suggestions into the revised manuscript. The main changes made include:

1. Providing more specific details about the methods

2. Expanding the theoretical discussion of plant trait influence on flammability and charcoal production across various fuel types.

3. Expanding the recommendations for future research on experiments examining the influence of plant traits and chemistry on fuel flammability and experiments conducted in the open flame and field settings.

We are aware that factors affecting charcoal production in field conditions can differ from controlled experimental settings. However, lots of these factors would be complicated to control and not all too realist. We are looking forward to seeing this manuscript published.

Your sincerely,

Angelica on behalf of coauthors

**Detailed response to both reviewers.**

***RC1***: 'Comment on egusphere-2023-1266', Anonymous Referee #1, 05 Aug 2023 reply (in red in annotated manuscript)

*Aim i) is presented well; ii) thresholds are only presented to a limited degree, can you make it clearer if thresholds are useful for these study sites and the limitations for applying them elsewhere? A word search shows that ''threshold'' is only used 3 times in the manuscript. iii) time should also be a factor here for combustion at temperature and is not presented in detail in the methods or the effect on the results. Can this be presented and explored further.*

R: i) Thank you;  ii) In the original version of the manuscript, we consistently used the term "cut-off values" throughout, with only a few instances where "threshold" was used interchangeably. We recognize that this interchange might have given the false impression of insufficient discussion regarding the threshold/cut-off values. Both the terms 'threshold' and 'cut-off' have been used in the literature to refer to interpretative boundaries for charcoal morphometrics, and the distinction between these terms is mainly semantic. To address this concern, please refer to sections 4.2 and 4.5, which contain an in-depth comparative discussion of the cut-off values across various fuel types and an exploration of limitations. iii) In the revised version of this manuscript, we have expanded upon the methods employed in our study. However, it is essential to note that once the fuel was placed in the muffle oven, we had limited control over the exact moment when fuel combustion occurred, as would be the case for all previous experimental charcoal production studies.

Specific comments

The comments about the abstract seem to seemingly be made without reading the full paper. Please see below our response to these comments.

*L43 burning experiments linking known plants to these metrics are limited - can you expand on the limitation(s) and the knowledge gap for open canopy ecosystems more here?*

R: The limitations of burning experiments in general and open ecosystems in particular have already been presented in the Introduction, as indicated in lines 87-100. In the abstract, we acknowledged limitations and emphasized the necessity for further studies to address them. To accommodate the reviewer comments below (L.98), we have extended our discussion to include taphonomy. Please see l. 95-101, l. 409-415.

*L44 along with selected Holocene charcoal assemblages - how many will be presented? A map or similar could be useful for presenting this information.*

R "This study presents novel analyses of laboratory-produced charcoal of 22 plant species from  the steppe regions of Eurasia (Romania and Russia), along with selected samples from three Holocene charcoal and pollen records from the same areas." see l. 39-41.

*L47 - of charcoal these plants produce - I suggest changing the language to be more careful about the burning of plants produces the charcoal.*

R: Thank you.

*L55 dry vs wet open ecosystems. - as these are relative terms associated with Precip/Evapo ratios, can you use other terms, for example aquatic/semiaquatic and terrestrial? Or non-aquatic? Earlier you used wetland gramminoids, which was more precise language.*

R: Thank you for pointing this out; we use grasses and wetland graminoids, or wetland vs. terrestrial ecosystems, in revising this manuscript.

*L58-60 long transport was also noted from observations in the field and included grass leaf char Pisaric, M.F., 2002. Long-distance transport of terrestrial plant material by convection resulting from forest fires. Journal of Paleolimnology, 28, pp.349-354. Courtney Mustaphi CJ, Vos HC, Marchant R, Beale C. 2022. Charcoal whirlwinds and post-fire observations in Serengeti National Park savannahs. Tanzania Journal of Science 48(2), 460–473.*

R: These references are added to those on l. 100-102

*L98 - consider to add ''and taphonomy'' with some references (among others)::*

*Scott, A.C., 2010. Charcoal recognition, taphonomy and uses in palaeoenvironmental analysis. Palaeogeography, Palaeoclimatology, Palaeoecology, 291(1-2), pp.11-39.Scott, A.C., Cripps, J.A., Collinson, M.E. and Nichols, G.J., 2000. The taphonomy of charcoal following a recent heathland fire and some implications for the interpretation of fossil charcoal deposits. Palaeogeography, Palaeoclimatology, Palaeoecology, 164(1-4), pp.1-31.Mustaphi, C.J.C., Vos, H.C., Marchant, R. and Beale, C., 2022. Charcoal Whirlwinds and Post-Fire Observations in Serengeti National Park Savannahs. Tanzania Journal of Science, 48(2), pp.460-473.*

R: We have extended the text to accommodate the references suggested and the problem of taphonomy in the Introduction (l. 93-99) and Discussion (l. 407-413).

*L62 - However, the relationship between woody charcoal and pollen may be more complex for trees, as their pollen can travel longer distances compared to woody charcoal. Can you reword to make sure you only intended atmospheric transport and not slope or water-bourne transport of woody charcoal.*

R: Done: "However, the relationship between woody charcoal and tree pollen may be more complex, as tree pollen can travel atmospherically longer distances compared to woody charcoal."

*L79 - remove ''inorganic''* R: Done

*L102 - laboratory-produced (muffle oven) charcoal - did you also compare the same fuels burned by open flame?*

R: No burns in open flame were conducted in this study. However, we touch on how the charcoal morphometrics of muffle ovens compare to those performed in open flame from literature see l-280-285 and concluding lines l-290-295: "The comparison of *L/W* ratio of charcoal particles produced in a muffle oven with those generated under open flames conditions reported in literature reveals slightly longer *L/W* values for particles produced in the open flames. Although this observation underscores the varying fragility of charcoal produced under different burning conditions, it also highlights that the relative proportion of the *L/W* ratio *between different fuel* types remain consistent. " l. 284-288.

*L107 - a database? Or only a list in a table? Can more information be added here? Like sample sizes (n values)?*

R: We have provided an extended text on the method on complied values on l. 144-153.

*L109 - open systems dominance - can this be clarified? Was the intention to mean the predominance of gramminoid fuels on the landscape? (catchment?) or the actual openness:closedness of the upper canopies?*

R: We have rephrased to: ''iii) thresholds in charcoal morphometrics such as *L/W*, *W/L*, and *A/P* ratios indicative of systems dominated by grasslands or a grass-tree mosaic such as steppe, savanna, forest-steppe, woodlands;''. See l. 109-110.

*L111-112 - the goal to provide tools for managers seems like a much further step from the 3 aims of the paper.*

R: Agree, we rephrased this to: "Ultimately, our approach aims to provide an accessible tool to understand fire regimes in temperate grass-dominated ecosystems''. L.120

*L116 - how does this sampling relate to the known plant richness and biodiversity? Or even abundances. Are these the common taxa? Please add more context to the field sampling, which is not presented.*

R: Added: ''These specimens are representative of the most prevalent plant taxa in the study area; however, this taxa list constitutes only a partial representation of common grasses and forbs. Many of these species are distributed in small patches due to the extensive conversion of natural steppe into agricultural fields..''. L. 118-1.121.

*L120 again what might be different between in situ, fully oxygenated burning and flame burning? And L123 - Can you add more methods detail here for people who may replicate or reproduce the study or apply elsewhere? How long at peak temperature, what was the ramp up duration for the oven? Time is an important factor for smouldering, how was time at temperature factored into the methods and results?*

R: We have expanded on the method used on l. 124-133; l.140-145. We do not know the time when the material was burnt. However, it is important to note that once the fuel was placed in the muffle oven, we had limited control over the exact moment when fuel combustion occurred, as would be the case for all previous experimental charcoal production studies. Since we have not done any open flame experiments, a thorough exploration of differences between the two types of measurements is beyond the scope of this manuscript. However, open flame experiments are not all realistic.

*L132 - is the width (W) always perpendicular to the L axis?*

R: Yes, always.

*L149 - did you mean typical diagnostic features of charcoal? See Hawthorne et al 2018 Quat Int Hawthorne D et al . 2018. Global Modern Charcoal Dataset (GMCD): a tool for exploring proxy-fire linkages and spatial patterns of biomass burning. Quaternary International 488, 3–17.*

R: "All plant material burnt had a typical charcoal appearance black, opaque, angular, planar (Whitlock and Larsen 2001) after being subjected to a temperature of 250°C". l. 172.

*L170 - area and perimeter in one 2D plane, what was done for conspicuously 3D particles?*

R: Yes, in 2D space. As for particles in 3D space, the L/W ratio should ideally be preserved. ''We measured the major (L) and minor (W) axes and surface area (A), and perimeter (P), of each particle following the algorithm of Feurdean (2021) and then calculated the aspect (*L/W; W/L*) and *A/P* ratios. A small number of identified charcoal particles were excluded from the quantification because they were not adequately identified. This misclassification was often due to their blurred aspect, partly overlapping particles, or not entirely within the picture frame''. L.138-144.

*L188 foliated? In the geologic textural sense? Or is this term used in botany also?*

R: In a botanical sense.

*L205 - is it worthwhile to explore some of the caveats and assumptions on how muffle burning only approximate real-world fires? There is some discussion at L219 that is still limited, can you also add what might be different and what could be considered for improved methodologies in further studies.*

R: We have slightly expanded the 4.1 section to also incorporate comments from the other reviewer, L.243-255, 262-266, as well recommendations for the future in the Conclusions l.460-466

*L226 - long term combustion through smouldering also reduces wood and grass to white ash. Mustaphi, C.J.C., Vos, H.C., Marchant, R. and Beale, C., 2022. Charcoal Whirlwinds and Post-Fire Observations in Serengeti National Park Savannahs. Tanzania Journal of Science, 48(2), pp.460-473.*

R: Reference included.

*L246 - what is meant by mechanical tissues? Consider adding a reference for nonspecialists.*

R: Resistance tissue.

*L256 - suggest replacing circularity with ''roundedness'' or something similar from particle shape descriptors.*

R: Circularity was replaced by roundness.

*L291 - it is difficult to understand what fires in dry grasslands within the reeds might mean? Could more be done to present these interpretations with photos from the field or heuristic diagrams?*

R: Revised: "At the Mangalia Herghelie site in Romania, samples with the highest aspect ratios (4.4-5.5) exhibit a considerable abundance of graminoid leaf morphologies (34-44%) and pollen from both dry (23-25%, Poaceae, Cereals) and wetland (9-23%, Cyperaceae, *Typha*) graminoids. Additionally, the stratigraphy at this site indicates the presence of peat with abundant remains of another wetland graminoid, *Phragmites,* which has a similar pollen morphology to Poaceae. Based on the evidence

above, we propose that fires likely occurred in both dry grassland areas and within the reed and sedge vegetation that developed around and on this site". L.333-339.

*L349 - see also Pisaric, M.F., 2002. Long-distance transport of terrestrial plant material by convection resulting from forest fires. Journal of Paleolimnology, 28, pp.349-354.*

R: Added

*L351 - This may have been argued by Belcher et al 2005, consider checking and citing. Belcher, C.M., Collinson, M.E. and Scott, A.C., 2005. Constraints on the thermal energy released from the Chicxulub impactor: new evidence from multi-method charcoal analysis. Journal of the Geological Society, 162(4), pp.591-602.*

R: Reference added.

*L351-353 consider citing the main decomposition method studies here for readers. One caveat being, those methods were first developed for ecosystems with dense mixed conifer dominated forests that have low frequency (100+ fire return intervals) and high intensity forest fires.*

R: "The decomposition methods were originally developed for North American forest ecosystems characterised by mixed conifer trees (Gavin et al., 2005; Higuera et al., 2009). These forests typically experience high intensity fires with fire return intervals exceeding 100 years. In contrast, grass fires, are characterised by low intensity and high frequency, although there is limited knowledge about fire frequency in temperate compared to tropical grasslands (Pausas et al., 2005; Feudean et al., 2018; Stevens et al., 2022)". L. 408-414.

*L363 - This suggests an under-representation of grass morphologies in the sedimentary charcoal record relative to forbs. Is this true for this ecosystem? This ratio of forbs to grasses on the local landscapes of the catchments? Or in general everywhere on grassy/herbaceous ecosystems?*

R: We have observed similar results: charcoal production is lower in grasses than forbs in Romanian and Russian grassland-dominated areas. This pattern is also reflected in boreal ecosystems (Feurdean, 2021; Pereboom et al., 2020). These consistent findings suggest the possibility of observing a lower prevalence of grass charcoal compared to forbs in broader grassland ecosystems or grassy/herbaceous ecosystems.

*L364 - is this finer distinction in the dimension of taxonomies, or plant parts, or taphonomy? Fire types? Vegetation structure? Can you conclude more closely to the data and interpretation that you have presented?*

R: We have clarified that charcoal morphologies offer a finer taxonomical and plant parts distinction of burnt vegetation than morphometrics. "Morphometric analysis indicates that graminoid charcoal particles are more elongated (4.0±2.5) than forbs (3.1±2.2); however, literature compilation shows some overlap with the global aspect ratio of wood (3.04±0.4). Nevertheless, *L/W* values above 3 (*W/L* below 0.40) may indicate predominantly herbaceous morphologies in temperate open ecosystems (steppe, forest steppe). "l. 436.440.

*L370 - the distributions overlap, how can this be further disentangled? Or is it not possible at present?*

We have revised this text to emphasize that charcoal morphologies can be valuable for distinguishing between forb and grass-charred particles and herbaceous and wood particles: ". As morphologies provide a more effective way of distinguishing charred particles from forbs, grasses, and wood, we recommend integrating these analyses to enhance distinctions between groups with overlapping morphometrics. " l. 438-440.

*L384 - this point was raised by Mustaphi and Pisaric 2014 Prog Phys Geog (already cited in manuscript) that the charcoal in sedimentary records needs to be observed site specifically before and requires an flexible and modifiable morphotype framework that can be reduced or expanded at any new site. At those study sites, the changing morphotype assemblage could not be disentangled between Hydroclimate change, fire type, and taphonomic mechanisms during the Holocene.*

R: We added the references above and the text to show that taphonomy mechanisms and hydrological conditions may need to be considered. "Hydroclimatic conditions and taphonomy have also been identified to contribute to these differences (Mustaphi and Pisaric, 2014)". l. 374-375.

*L399 - the methods for the review and literature database are not presented*

R: Done, see our response above to L .107.

*Throughout manuscript - The use of italics for taxonomic names varies in the text, tables and figures.*

R: We have italicized the taxonomic names also in figures.

*Figure 1 caption - unclear what ''their origins are'' is intended to convey? Geographic origin? Plant part?*

R: The full names of plant species burnt, and their geographic origins are presented in Table 1.

FIGURE                                                                                                  3

*Check taxonomic name spellings throughout, for example final panel of Figure 3 misspelt 'Agropiron critatum (l)' (Line570)*

R: Corrected.

*Deformation during ignition is a subject that is rarely discussed in the scientific literature as a major taphonomic process. Is some of the curvature featured in the grass charcoal of Figure 3 reflecting some modification during ignition?*

R: We do not know why, but some grasses often had curvature features.

*Table 1 - it might be worth stating what taxonomic name system is being used for the synonyms and how there is a slash used on some taxa.*

*Table 1- cf. and sp. Are often italicised by accident.*

R: We have retained a single taxonomic name.

*Table 3 - How complete is this literature review? In the methods section it was not introduced how and when the literature review was done or if it is ad hoc and non-exhaustive.*

R: Please see our full response to L.207

*L683 - table 3 caption - pluralise ''parentheses''*

R: Done

*Table S2 - what are the units for Mass retained?*

R: Percentages

*Can you provide n values for measurements? How many measurements were included in the ratios? Readers can derive it from Supplement S1 but it would be useful to see it in the Figures or captions.*

R: n values added in the figure 2.

*File S1 column D and L and T etc header, misspelt ''lenght''*

*Length*

R: Corrected.

*L684 - trunks and twigs - are you certain of the woody anatomy that was burned? Or is this an assumption? Twigs in some contexts can be interpreted as specifically meaning the new woody growth in the most recent growing season.*

R: As we included only results of known plant material, we assumed that the twigs had woody anatomy.

**Reviewer 2 General comment on contribution** *(in blue in annotated manuscript)*

*The submitted manuscript "Charcoal morphologies and morphometrics of a Eurasian grass-dominated system for robust interpretation of past fuel and fire type" is a welcome addition to the growing paleofire literature from grass-dominated ecosystems. These ecosystems are poorly understood because the charcoal proxy biases preservation of fine fuels, and also, there are still issues with reconstructing fires as some charcoal size-classes can be rare in these systems that experience frequent fires. It is therefore*

*pleasing that the authors explored fuel characteristics of the dynamic herbaceous layer (i.e., grasses and forbs) to understand it's potential to produce charcoal, and the morphological characteristics of the charcoal produced. In this regard, the paper makes a key contribution: grasses produce less charcoal compared to forbs, and that elongation ratios are difficult to intepret.*

**Issues of concern**

*While I found the pre-print easy to read and presenting current palaeofire knowledge based on the references, I have key objections based on other ecological knowledge about fire that the authors would have been privy to, and that would have added to the work.*

*1.The authors aim to produce robust results on fine fuel characteristics and charcoal production but have erred by not considering key variables that affect charcoal production: flammabaility, which can be reduced to combustibility, ignitability, and sustainability of flame. Key references to consider here are Simpson et al. (2016)'s work. And she has done lot of groundwork flammability of grasses from an eco-evolutionary perspective, I recommend reading Simpson's work, also check Pausas et al (2017) and Bond and Keeley (2005). The consensus is that traits **matter** as fuel amount (depending on plant size) and fuel moisture content are eco-evolutionarily determined. And since the herbaceous layer is fast-flammable (see Pausas) and easily ashed depending on fuel curing and other fire weather characteristics, it seems unreasonable to measure charcoal production of dry fuels as these rarely meet field conditions, and would generally produce less charcoal because of more complete combustion. Perhaps make note of this, and see how Simpson got around problem of flammability.*

*2.This is related to the above point, but I will stress it separately. According to Simpson, biomass density is not a significant factor for flammability, and by reasoning, charcoal production. This can take away much of the discussion points you made in first section of discussion. However, biomass density only matters when it is related to grass size, for example, tall grasses and reeds have higher biomass density and lignin. And for reed grasses, we know that they are less flammable as they dry out less frequently. And as you found out forbs.*

R: Thank you for the pertinent points. We agree with them but acknowledge that most would be extremely difficult to control. Nevertheless, in revising this manuscript, we add text on the role of a range of plant traits important for flammability (biomass quantity, density, moisture content, leaf-to-area volume ratio, among others) in charcoal production, fragility, and dispersal. We also added that areas with higher moisture levels, such as wetlands, are expected to exhibit resistance to combustion due to elevated soil and fuel moisture content. In future experimental laboratory research on charcoal production, we also recommend incorporating a broader range of fuel moisture conditions that closely resemble real-world scenarios. Please see our expanded responses on l. 227-230; 244-254, 262-260; 276-281; 284-290; 454-462.

*The authors have not imagined how charcoal production factors in field settings would differ from experimental production. I think this is imprtant for the field to progress and for the design of future studies. For example, in the field, fires spread at different rates, consuming from homogenous to heteregoenius fuels, with rates presumably influencing charcoal production, charcoal fragility, and subsequent morphometry. I know many studies have focused on charcoal aerial dispersal from fires and charcoal production from individual plants but not the biotic resistance that produces charcoal. This biotic resistance can be expressed as flammability/ignitiability. For example, I expect*

*the resistance to be higher towards the wetland because of higher soil and fuel moisture, meaning most charcoal is locally produced. This will inform the way I interpret multiple proxies--for example, forb abundance, grass abundance, reed abundance, and C/N ratios related to lignin of biomass density. And aother proxies like phytoliths.*

R: We are aware that factors affecting charcoal production in field conditions can differ significantly from controlled experimental settings, impacting charcoal production, fragility, and morphometry. Investigations, including chemical analyses such as nitrogen, phosphorus, and tannin content of fuel, can enhance the evaluation of their influence on charcoal production. Plant architecture has a significant role in flammability; therefore, considering combustion at the whole plant scale rather than focusing on small plant parts can be another improvement. However, as stated above, lots of these factors would be extremely difficult to control. See our lines suggested above.

*4.Perhaps the identification of fuel type is a very lofty goal to achieve using charcoal metrics produced under different fuel conditions, we may be more likely to understand general fuel characteristics of particular firs compared with fine detail about specific fuel types.*

R Our primary objective in employing charcoal morphologies and morphometrics is to obtain the most accurate approximation of fuel composition and characteristics attainable within these constraints.

***References*** *Simpson: 10.1111/1365-2745.12503; Pausas: https://doi.org/10.1111/1365-2745.12691 ;Bond and Keeley: https://doi.org/10.1016/j.tree.2005.04.02*
R: references included.

**Minor editorial details**

*I prefer growth-forms to growth-habits:*

R We have used growth form in the revised paper.

*line 64: compared with woody or grass/herb fuel?*

R: ''However, the relationship between woody charcoal and tree pollen may be more complex, as tree pollen can travel atmospherically longer distances compared to woody charcoal''. L. 61-63.

*; line 73: ecological to evolutionary; )*

R: ---at evolutionary to ecological timescales (Bond et al., 2004). L. 69

*line 75: composition of what*

R: "Consequently, little is known about the natural occurrence and intensity of fires in relation to climate, biomass amount and vegetation composition in these ecosystems". L 70-71.

***Given the issues identified, perhaps***

*In table 1, mention if the grasses are tall or short, and whether they are C3 or C4.*

R: List of plants burned from Dobrogea, the Black Sea, Romania (Ro) and Konoplyanka, Trans-Urals, Russia (Ru). All grasses are C3.

---

## Author Response (AR2)

13.10.2023 Frankfurt am Main

Dear Editor-in-Chief and associate editor, Petr Kuneš

Thank you for your comments on our revised manuscript: *Charcoal morphologies and morphometrics of a Eurasian grass-dominated system for robust interpretation of past fuel and fire type.* We have carefully considered and incorporated your suggestions into the revised manuscript.

*l.70 - "open systems" - perhaps it is better to use "open landscape" or "open ecosystems"*
*R:* Done: "One of the largest extents of open ecosystems is in Eurasia, which has been heavily impacted by human activities for millennia". L 70.

*l.82 - instead of "characteristics of the plants" use "plant traits"*
R: Done: "These keys can help link charcoal particles to plants traits, such as plant types (moss, graminoids, forbs, shrubs/trees) or plant parts (stems, branches, roots, leaves, wood)." l. 82-83.

*Material and Methods - similar to rev #1, I find it useful to have a map, you could indicate the position of your study areas (nowhere to be found) and also the positions of forest zones (l.123)*
R: Map presented as figure 1.: **Figure 1. T**he distribution of open ecosystems (steppes/grasslands and forest-steppes) in Eurasia. Mangalia Herghelie, Black Sea, SE Romania (1), and Tom and SH sites, Konoplyanka, Trans-Urals, Russia (2), are located in the grassland ecosystems. l.638.
The geographical coordinates of these sites are also provide in the Methods: ''To demonstrate the applicability of experimental charcoal morphologies and morphometrics for the identification of fuel burnt in sedimentary records, we randomly selected five Holocene samples from a core taken from Mangalia Herghelie wetland, Romania (43.838056 N, 28.583333 E), and six samples from wetland sites near the Karagaily-Ayat river (profile Tom 52.864648 N, 60.222420 E and profile SH 52.858475 N, 60.226214 E), where our plant material samples were collected (Fig. 1). L. 157-162.
The existing maps presented the open ecosystems together (grasslands and forest-steppe), thus so does our Fig. 1. For details on the location of the forest-steppe, the readers will need to rely on the information in the text and the geographical coordinates we provide.

*l.138 - It is not clear to me how the particles were "automatically detected"? Is it the feature of the camera used?*
*R:* The particles on these photographs were automatically detected following an algorithm described in Feurdean (2021) see also figure below. The revised text reads "Photographs of charcoal particles were manually taken at 4X magnification with a digital camera (KERN ODC 241 tablet camera. The charcoal particles and morphometric measurements, including the major (L) and minor (W) axes surface area (A), and perimeter (P) for individual charred particles larger than 150 μm, were automatically determined from these photographs using the algorithm of Feurdean (2021). Subsequently, we calculated the aspect (*L/W; W/L*) and *A/P* ratios). These metrics were applied on more than 150 charcoal particles (range between 41 and 508), per sample; the lower number of measurements generally corresponds to samples burnt at high temperatures, where particles were more susceptible to breakage or partial ashing". l. 136-142.

The figure below shows a picture with automatic detection of charcoal particles in photographs. Afterwards, the algorithm calculates the major (L) and minor (W) axes surface area (A), and perimeter (P).

[Figure]

*l.335 - Phragmites belongs to the Poaceae family and its pollen grain is, therefore Poaceae-type. Please reformulate this sentence.*

R: Done: '' Additionally, the stratigraphy at this site indicates the presence of peat with abundant remains of wetland graminoid *Phragmites,* which belongs to the Poaceae family, and its pollen morphology is, therefore, Poaceae-type". l. 336-338.

In addition, Data availability: Apart of storying these measurements in the SI, we are also in process of storing it FAIR and OPEN access. At the poof stage we will likely be in the position to provide a DOI for the data storage. The text now reads: '**Data availability:** All raw measurements of length, width, area, perimeter, and ratios of *L/W, W/L*, and *A/P* for all taxa and temperatures are presented in File S1. This dataset is going to e stored at the GFZ Data Services under a CC BY 4.0 International License and will soon be assigned Digital Object Identifiers (DOI). https://dataservices.gfz-potsdam.de/portal/about.html.

Kind regards,
Angelica Feurdean, on behalf of the coauthors

---

## Author Response (AR3)

30.10.2023 Frankfurt am Main

Dear Copernicus production team,

*Regarding your figure 1: for the next revision, please check if your figures containing maps/aerial images require a copyright statement/image credit and add it to the figures (or captions) (https://publications.copernicus.org/for_authors/manuscript_preparation.html#mapsaerials). If these figures were entirely created by the authors, there is no need to add a copyright statement or credit. In that case it is important that you confirm this explicitly by email.*

R: The following text was added to the caption of Fig. 1: "**Figure 1. T**he distribution of open ecosystems (steppes/grasslands and forest-steppes) in Eurasia. Mangalia Herghelie, Black Sea, SE Romania (1), and Tom and SH sites, Konoplyanka, Trans-Urals, Russia (2), are located in the steppe ecosystems. The source image is a product of NASA's Blue Marble Project public domain'. L. 639.

Kind Regards
Angelica